# Time-domain R-PDLF NMR for molecular structure determination in complex lipid membranes

Anika Wurl[1], Kay Saalwächter[1], and Tiago Mendes Ferreira[1]

[1]NMR group, Institute for Physics, Martin Luther University Halle-Wittenberg, Halle (Saale), Germany

**Correspondence:** Tiago Mendes Ferreira (tiago.ferreira@physik.uni-halle.de)

**Abstract.** Proton-detected local field (PDLF) NMR spectroscopy, using magic-angle spinning and dipolar recoupling, is presently the most powerful experimental technique for obtaining atomistic structural information from small molecules undergoing anisotropic motion. Common examples include peptides, drugs, or lipids in model membranes and molecules that form liquid crystals. The measurements on complex systems are however compromised by the larger number of transients required. Retaining sufficient spectral quality in the direct dimension requires that the indirect time-domain modulation becomes too short for yielding dipolar splittings in the frequency domain. In such cases the dipolar couplings can be obtained by fitting the experimental data, however ideal models often fail to fit PDLF data properly due to effects of RF spatial inhomogeneity. Here, we demonstrate that by accounting for RF spatial inhomogeneity in the modelling of R-symmetry based PDLF NMR experiments, the fitting accuracy is improved, facilitating the analysis of the experimental data. In comparison to the analysis of dipolar splittings without any fitting procedure, the accurate modelling of PDLF measurements makes possible three important improvements: the use of shorter experiments that enable to investigate samples with a higher level of complexity; the measurement of C–H bond order parameters with smaller magnitudes $|S_{\mathrm{CH}}|$ and of smaller variations of $|S_{\mathrm{CH}}|$ caused by perturbations of the system; and the determination of $|S_{\mathrm{CH}}|$ values with small differences from distinct sites having the same chemical shift. The increase in fitting accuracy is demonstrated by comparison with $^2$H NMR quadrupolar echo experiments on mixtures of deuterated and non-deuterated dimyristoylphosphatidylcholine (DMPC) and with 1-palmitoyl-2-oleoyl-$sn$-glycero-3-phosphoethanolamine (POPE) membranes. Accurate modelling of PDLF NMR experiments is highly useful for investigating complex membrane systems. This is exemplified by application of the proposed fitting procedure for the characterisation of membranes composed of a brain lipid extract with many distinct lipid types.

## 1 Introduction

The methodology for characterizing molecular structure in biological systems has been advancing rapidly over recent years (Cheng, 2018; Gauto et al., 2019; Wu and Lander, 2020). Among the various experimental techniques available, solid-state NMR spectroscopy provides the most powerful methods for investigating the molecular structure of smaller molecules (<30 kDa) undergoing anisotropic motion in membranes such as lipids, peptides and membrane proteins (Andersson et al., 2017; Löser et al., 2018; Bacle et al., 2021; Zerweck et al., 2017; Aisenbrey et al., 2019; Cady et al., 2007; Park et al., 2012; Mandala et al., 2020; Umegawa et al., 2018; Mandala et al., 2018).

Of these techniques, researchers have consistently used $^2$H NMR spectroscopy since the 1970s for the molecular structural characterization of both lipids and peptides (Seelig, 1977; Davis, 1983; Strandberg and Ulrich, 2004; Leftin and Brown, 2011). The popularity of this method is due to both its simplicity and the high accuracy of the experiments. However, it requires specific $^2$H isotopic labelling for site resolution. This requirement severely limits its application to the investigation of biological extracts or complex model membranes that mimic biological systems. Alternatives to $^2$H NMR are separated local field (SLF) 2D NMR experiments that make use of $^1$H–$^{13}$C heteronuclear dipolar recoupling during the indirect time (Hester et al., 1976). This technique is very advantageous compared to $^2$H NMR because SLF delivers the same type of information with $^{13}$C chemical shift selectivity and does not require isotopic labelling. These SLF $^1$H–$^{13}$C heteronuclear dipolar recoupling NMR techniques fall into two groups, carbon-detected local field (CDLF) and proton-detected local field (PDLF) experiments with the latter having enhanced resolution (Nakai and Terao, 1992; Schmidt-Rohr et al., 1994; Bärenwald et al., 2016). Here, we discuss only PDLF techniques which provide C–H bond site resolution and are therefore directly comparable to $^2$H NMR.

A number of PDLF pulse sequences have been implemented to investigate lipid membranes as well as other systems (Griffin, 1998; De Paëpe, 2012; Molugu et al., 2017). The main difference between the distinct PDLF sequences reported are the heteronuclear dipolar recoupling blocks used to counteract the effect of magic angle spinning (MAS) on the anisotropic Hamiltonian terms during the indirect dimension. Several recoupling pulse sequences have been designed for this purpose such as in REDOR (Gullion, 1998), DROSS (Gross et al., 1997; Leftin et al., 2014), R-symmetry based pulse sequences (Levitt, 2007; Dvinskikh et al., 2004; Hou et al., 2011) and others (De Paëpe, 2012). In recent years, a R-symmetry based PDLF sequence designed by Dvinskikh and coworkers (Dvinskikh et al., 2004) dubbed R-PDLF (incorporating $R18_1^7$ recoupling blocks) , has been frequently used to successfully investigate lipid membranes (Dvinskikh et al., 2005; Ferreira et al., 2013; Löser et al., 2018; Bacle et al., 2021; Fridolf et al., 2022). The R-symmetry sequences are particularly advantageous since they enable to simultaneously recouple the heteronuclear and decouple the homonuclear dipolar interactions. Moreover, R-symmetry recoupling can be applied directly to an $IS$ spin ensemble in thermal equilibrium with no need of generating transverse magnetization before the recoupling blocks and therefore avoiding potential $T_2$ losses during the recoupling period, i.e. increasing sensitivity. It is well known that, as with other dipolar recoupling pulse sequences, R-symmetry recoupling is rather sensitive to RF imperfections (Nishimura et al., 2001; Schanda et al., 2011) and therefore to the RF spatial inhomogeneity across the sample investigated – one unavoidable and unique feature of the MAS probe used to perform experiments (Tošner et al., 2017). To minimize this problem, windowed R-symmetry sequences have been proposed that are not as sensitive to RF inhomogeneity as the original windowless R-symmetry sequences (Gansmüller et al., 2013; Lu et al., 2016). These windowed versions, however, require using higher RF radiofrequency fields which is a practical bottleneck due to hardware limitations.

In general, the strategy for developing better dipolar recoupling methods has been to minimize the sensitivity to RF spatial inhomogeneity, by using/designing dipolar recoupling sequences that are less sensitive to RF imperfections and reducing the sample studied to a narrow volume, as well as using recoupling pulse sequences with a high scaling factor that enable to measure smaller dipolar couplings (Dvinskikh et al., 2005; Chevelkov et al., 2009; Schanda et al., 2011; Gansmüller et al., 2013; Lu et al., 2016; Asami and Reif, 2017; Jain et al., 2019; Nimerovsky and Soutar, 2020). This last point is of significant importance, since higher scaling factors enable to obtain dipolar splittings in the indirect frequency domain with a reduced

maximum time in the indirect dimension. PDLF experiments have been mostly analysed by "reading off" the dipolar splittings obtained in the indirect frequency domain of the 2D spectra. In contrast to the standard analysis of REDOR or DIPSHIFT dipolar evolutions (Gullion, 1998; deAzevedo et al., 2008), PDLF measurements have been rarely analysed by employing fitting procedures. The few TD analysis attempts reported assumed samples under a homogeneous RF field (Gross et al., 1997; Gansmüller et al., 2013; Fridolf et al., 2022), however, it is well known that dipolar recoupling sequences are highly sensitive to RF inhomogeneity (Schanda et al., 2011).

Here, we present an analysis procedure that enables to increase the accuracy and applicability of R-symmetry based PDLF NMR through R-PDLF numerical simulations that take into account the RF spatial inhomogeneity of the probe used explicitly, i.e., rather than trying to minimize the effect of RF spatial inhomogeneity we include it in the NMR simulations used to fit the experimental measurements. The use of RF inhomogeneity for data analysis has been done recently for investigating protein molecular structure (Xue et al., 2022). Here we apply this strategy to analyse R-PDLF experiments that were proven suitable to investigate molecular structure in lipid membranes. The novelty of the procedure presented is solely in the analysis of the experimental data, no special hardware or different setup of the experiment is required. We tested the methodology on a DMPC/DMPCd sample to allow comparison of the $^1$H-$^{13}$C dipolar couplings determined by a R18$_1^7$-type PDLF sequence (R-PDLF (Dvinskikh et al., 2004, 2005)) with the C–D bond order parameters determined from $^2$H quadrupolar couplings. We then implemented such an analysis procedure to investigate a system of complex lipid membranes composed of a brain lipid extract. This enabled us to measure several dipolar couplings from the membrane hydrophilic layer composed of many types of distinct phospholipid headgroups. We believe that this strategy is highly useful for investigating both model systems and complex membrane systems with a low signal to noise ratio such as biological lipid extracts. In addition it has the potential for investigating in-cell molecular structures.

## 2   Methodological Framework

The typical approach for determining C–H bond order parameter magnitudes with PDLF dipolar recoupling NMR is by performing a Fourier transform,

$$I(\omega_1) = \int\limits_0^{+\infty} s(t_1)\exp(-i\omega_1 t_1)dt_1 \tag{1}$$

of the time-domain recoupled dipolar modulation over the indirect dimension, $s(t_1)$, and reading-off the splitting(s) of the resulting spectral dipolar line shape (Dvinskikh et al., 2004; Gross et al., 1997). If the scaling factor, $\kappa$, for the particular pulse sequence under use is known, determining the magnitude of a C–H bond order parameter from the splitting(s) observed is straightforward by using

$$|S_{\mathrm{CH}}| = \kappa^{-1}\frac{\Delta\nu}{d_{\mathrm{CH}}} = \left|\frac{1}{2}\langle 3\cos^2\theta - 1\rangle\right| \tag{2}$$

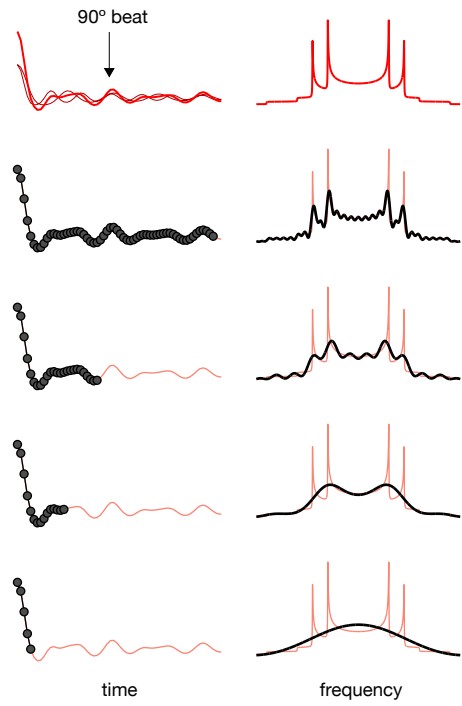

**Figure 1.** The effect of using a limited number of time-domain points to describe a superposition of two Pake patterns with distinct splittings.

where $d_{CH}$ is the magnitude of the rigid dipolar coupling for a static C–H bond equal to $\approx 22$ kHz, the angle $\theta$ is between the C–H bond and the symmetry axis of its uniaxial motion (in case of lipid membranes this is the bilayer normal), and the angular brackets denote a time average over a time interval up to approximately the inverse of the splitting measured.

For a dipolar modulation corresponding to a single C−H bond order parameter and acquired under optimal experimental conditions, the use of equation 2 allows accurate values to be obtained provided that the digitised signal, $s(t_1)$, extends to a time longer than the reciprocal of the scaled dipolar coupling frequency (e.g. 4 times the reciprocal gives approximately 10% error). If $s(t_1)$ is instead a superposition of two distinct contributions with different $S_{CH}$, for resolving the two components in the spectrum, the signal acquisition must be, at least, longer than half of a beat period defined by the frequencies of the crystallite orientations responsible for the two splittings (Lindon and Ferrige, 1980). This is illustrated in Figure 1 for a superposition of two Pake patterns. Therefore, both for single-component dipolar modulations with low order parameters and for two-component dipolar modulations with a small frequency difference between the two components, a large number of indirect dimension points is needed in 2D dipolar recoupling experiments which severely increases the experimental time. Such requisite is especially difficult to achieve for samples that require acquisition of a large number of transients in the direct dimension. Moreover, there is a maximum limit to the number of indirect dimension points acquired both due to RF heating of samples and to maintain the integrity of the NMR probe.

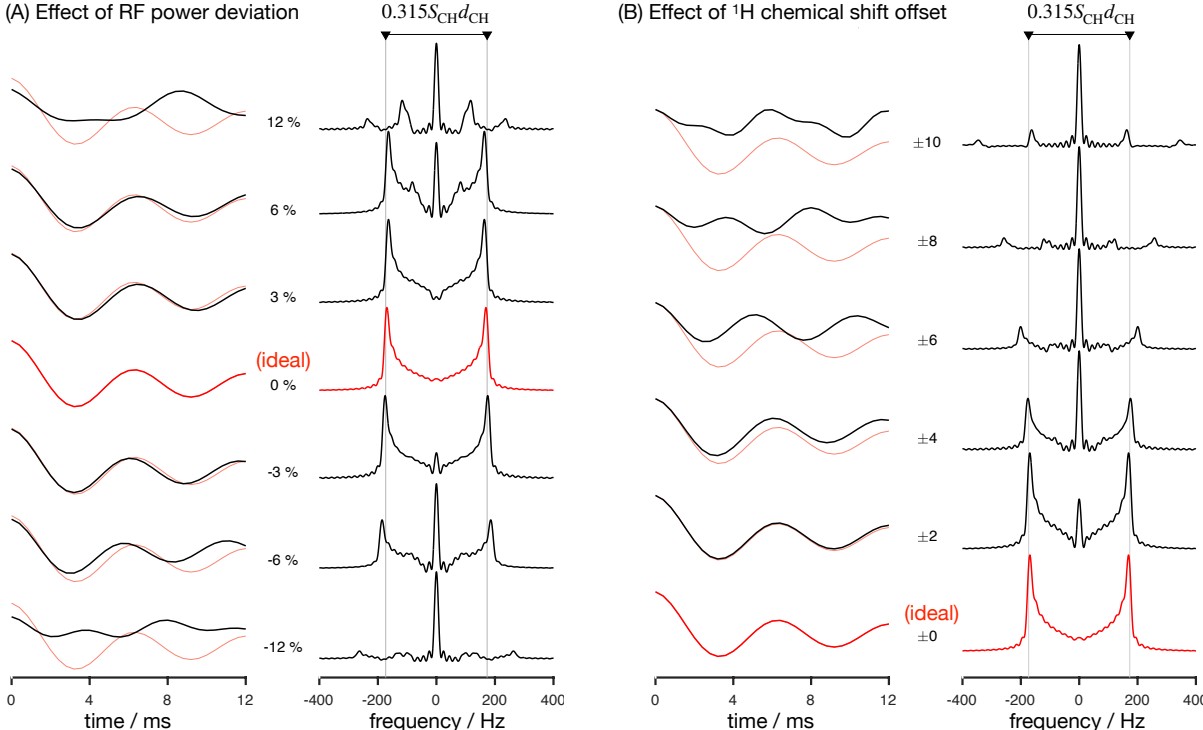

**Figure 2.** Numerical simulations of the R-PDLF pulse sequence for a fixed dipolar coupling displaying the effects of RF nutation frequency deviation from ideal settings (left) and $^1$H chemical shift offset (right) given in ppm units. The solid red and faint red curves correspond to the dipolar modulation with ideal settings which yield a dipolar splitting equal to $0.315 S_{\mathrm{CH}} d_{\mathrm{CH}}$. Negative and positive chemical offset shifts with the same magnitude have exactly the same effect. All simulations in this work were performed for a $^1$H Larmor frequency of 400 MHz.

Analysing the measured data with a fitting model (in either the time or frequency domains) can be used to circumvent the limitations outlined above. The practical bottleneck for such an analysis is the RF inhomogeneity across the sample, intrinsic to the majority of experimental setups (Tošner et al., 2017), combined with the sensitivity of recoupling pulse sequences to RF strength accuracy and the effect of chemical shift offsets (Schanda et al., 2011; Lu et al., 2016). Figure 2 illustrates such dependencies for the R-PDLF pulse sequence (Dvinskikh et al., 2004), showing a non-linear dependence of the dipolar

modulation on the RF pulse power level deviation from the ideal value and on the $^1$H chemical shift offset. The effect of RF miscalibration becomes rather pronounced at RF frequencies higher or lower than $6\%$ of the ideal frequency. This means that in typical experiments, for which the RF spatial inhomogeneity profile may reach $80\%$ of the maximum value in the outer parts of the sample (Tošner et al., 2017), one should expect deviations from the ideal dipolar lineshape. The effect of the $^1$H chemical shift offset is almost negligible up to 4 ppm above which large deviations from the ideal behaviour occur. For

achieving optimal resolution in an R-PDLF experiment the chemical shift offset should therefore be no more than 4 ppm at a magnetic field inducing a Larmor frequency for $^1$H equal to 400 MHz.

The outcome of an experiment is always the sum over all the detectable spatially distributed sample volumes. Therefore the experimental data measured with an R-PDLF experiment is a sum over dipolar modulations, each modulation with a characteristic lineshape that depends on the local RF field. To simulate realistic R-PDLF experiments accounting for RF inhomogeneity, we first measured the RF inhomogeneity in the probe used in this work by the method suggested by Odedra and Wimperis (Odedra and Wimperis, 2013) (details given as SI). The simulation data was then generated by integrating over R-PDLF NMR simulations that covered the RF spatial inhomogeneity measured. This was done by composing a MATLAB script with SIMPSON call-outs enabling to build an R-PDLF simulation database with realistic modulations for a range of $S_{CH}$ values. The database generated included the effect of chemical shift offsets and RF miscalibration with or without a built in RF spatial inhomogeneity profile. The MATLAB/SIMPSON files and the NMR database generated are available as open data (https://github.com/tfmFerreira/inhomogeneous-rf-nmr.git).

The result of accounting for RF inhomogeneity in the R-PDLF simulations is exemplified in Figure 3 for a two-component dipolar modulation, consisting of two distinct C–H bond order parameters of 0.03 and 0.04. The main effect of including RF spatial inhomogeneity in the simulation is the damping of the signal and the non-zero long time average of the modulation which gives rise to a middle peak in the dipolar spectrum as also reported previously by Polenova and coworkers (Lu et al., 2016). Due to the limited number of points used, the distinct dipolar couplings are not visible in either dipolar spectrum of the two simulations. Figure 3D shows the result of adding random noise to the simulated data in Figure 3C and fitting the data either with an ideal model or by taking the RF spatial inhomogeneity into account. The fit performed with NMR simulations having ideal settings gave values of 0.034 and 0.094, thus precluding the use of such fitting procedure for accurate analysis under such conditions. In contrast, the fit performed based on NMR simulations accounting for RF spatial inhomogeneity matches the data almost perfectly - as expected since these simulations were used to generate the original data - giving order parameters values of 0.028 and 0.040, very close to the original values 0.03 and 0.04 used. This suggests that, if the (coil dependent) RF spatial inhomogeneity across the sample is measured with reasonable accuracy, it would be possible to simulate a set of realistic data for the complete range of dipolar couplings (0-22 kHz for $^1$H-$^{13}$C). This would better describe the experimental measurements and therefore enable a more accurate analysis.

## 2.1  Results and Discussion

## 2.2  R-PDLF fits including RF inhomogeneity enable higher accuracy and shorter experiments

From the description shown in the preceding section it is clear that RF spatial inhomogeneity affects the dipolar modulation in R-PDLF experiments. Therefore, for accurate fits of experimental data, this effect needs to be taken into account. In this and the following section we demonstrate that accounting for the effect of RF inhomogeneity in time-domain fits of experimental data indeed leads to highly accurate fits of the experimental data and consequently to a considerable improvement of the accuracy of the C–H bond order parameters determined. Instead of in the time domain, the fitting could also be performed in the frequency domain, i.e. by performing the Fourier transform of the experimental data in the indirect dimension and by then fitting this data with the Fourier transform of the numerical simulations. Here, we simply use time domain fits and avoid performing

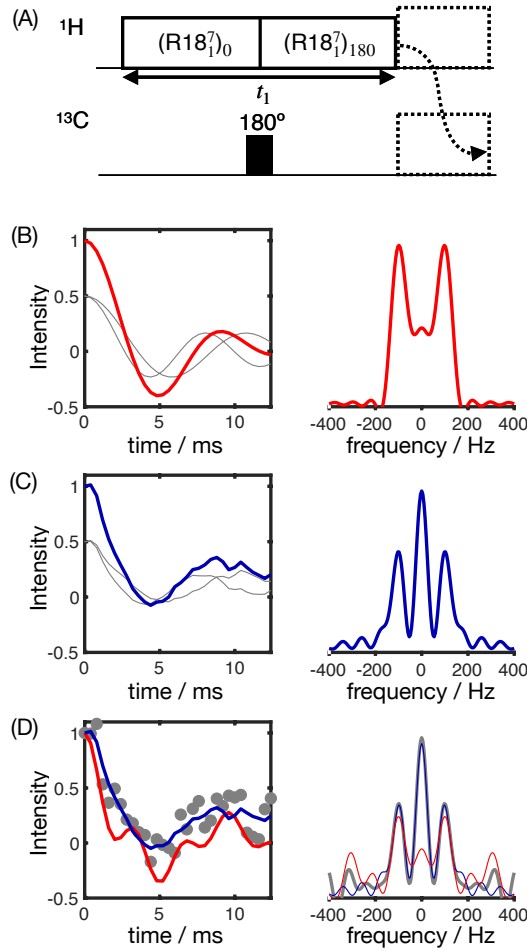

**Figure 3.** Simulation of the effect of RF spatial inhomogeneity on R-PDLF dipolar modulation and illustration of the advantage of time-domain over frequency-domain analysis of the data. (A) R-PDLF pulse sequence used in this work (Dvinskikh et al., 2004). (B) Ideal R-PDLF dipolar modulation with two components having C−H bond order parameters equal to 0.03 and 0.04 (grey lines display the individual components used). In the frequency domain, the two splittings can not be distinguished, due to the limited number of points used in the time domain. (C) The same as in (B) but including the effect of RF spatial inhomogeneity taken as a Gaussian distribution that reflects the RF inhomogeneity of the coil used in this work (details in the SI). (D) Simulation of an experimental result with low signal-to-noise ratio by addition of random noise to the simulated curve in (C).On the bottom plot, the curves show the result of fitting the data with two components using numerical simulations performed with ideal settings (red) and by accounting for the RF spatial inhomogeneity profile (blue).

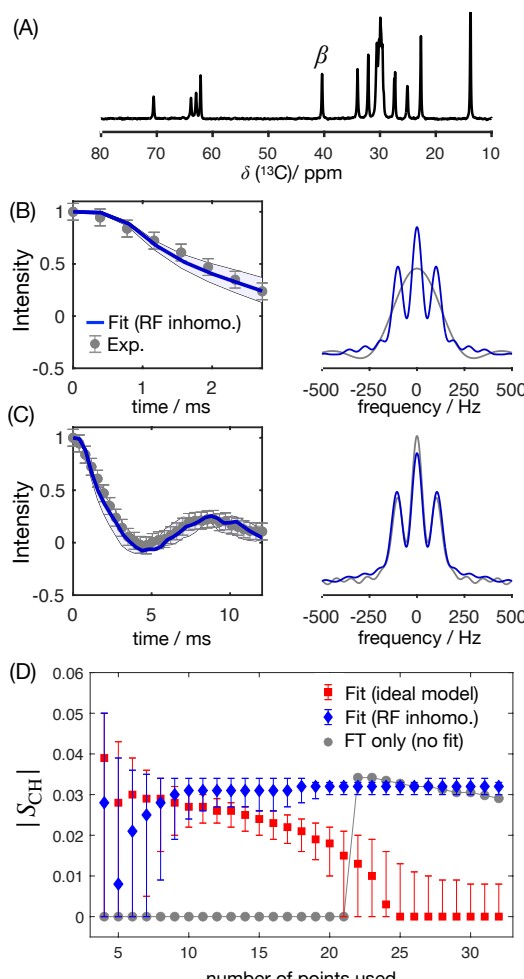

**Figure 4.** The proposed method applied to a sample of POPE MLVs. (A) $^{13}$C rINEPT spectrum of the POPE MLVs. (B) Time-domain (TD) fit accounting for RF inhomogeneity of the dipolar dimension of a R-PDLF experiment with a total of 8 points (thick solid line). The experimental dipolar spectrum is shown on the right together with the fit prediction of the experimental dipolar spectrum (using a total of 32 time-domain points). The thin lines show the time-domain curves that correspond to the error limits as defined in the SI. (C) The same as in (B) but fitting a total of 32 experimental points in the indirect dimension. A single-exponential decay of 14 Hz was multiplied to the simulated data. (D) Order parameter estimated by the time-domain fit including RF inhomogeneity (blue), with the ideal model (red), and calculated from the dipolar splitting in the Fourier transform of the experimental data (gray) as function of the total number of points in the indirect dimension used.

the additional step of the Fourier transform which is unnecessary and has an extra computational time cost. Figure 4 shows the application of this time-domain analysis to a sample of 1-palmitoyl-2-oleoyl-*sn*-glycero-3-phosphoethanolamine (POPE) MLVs, illustrating the analysis method on the $\beta$ carbon of the POPE headgroup. POPE is one of the most abundant lipids

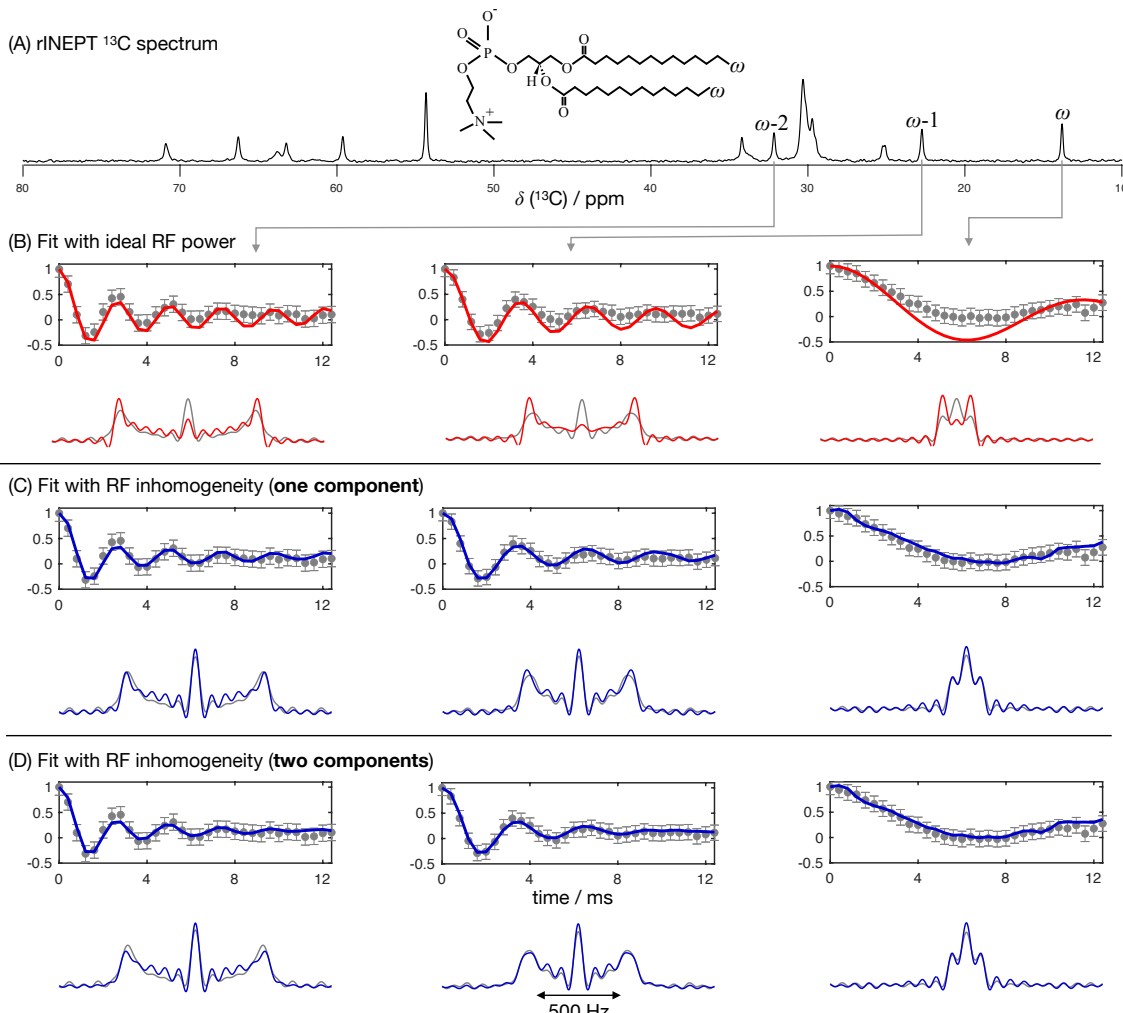

**Figure 5.** The proposed methodology applied to a sample of a DMPC/DMPCd54 liquid crystalline system. (A) $^{13}$C rINEPT spectrum and labels used to identify different carbons. (B) Time-domain fits neglecting the RF inhomogeneity across the sample. (C) Time-domain fits accounting for the RF inhomogeneity across the sample using one single dipolar coupling as fit parameter and neglecting any possible relaxation during $t_1$. (D) The same as in (C) but using two dipolar couplings as fit parameters assuming a two-component dipolar modulation.

in cellular membranes. It has recently been investigated with R-PDLF NMR in the context of the NMRlipids project. Such project aims to resolve the atomistic molecular structure of the most abundant lipids in nature. To this end, the project uses a

155 combination of solid-state NMR experiments and all-atom molecular dynamics simulations (Bacle et al., 2021). Using only the first 8 points along the indirect dimension, the time-domain curve fitted to the experimental dipolar modulation measured for the $\beta$ carbons gives an estimation of $S_{\mathrm{CH}} = 0.028_{0.009}^{0.034}$ where the superscript and subscript are upper and lower limits as defined in the Supplementary Information. Such fit enables to predict the experimental dipolar spectrum that one would obtain

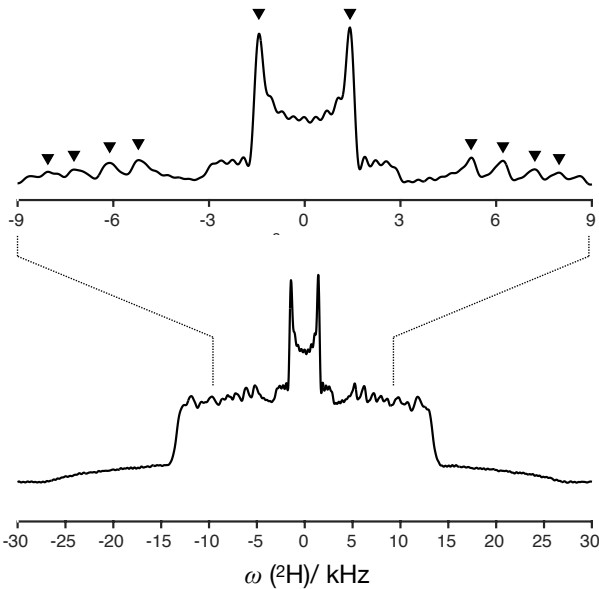

**Figure 6.** $^2$H NMR spectrum of the DMPC/DMPCd54 liquid crystalline system and symbols showing the positions used to determine order parameters.

using a higher number of indirect dimension points as shown in Figures 4B and 4C. Figure 4D shows how the time-domain
fit procedure enables to estimate an accurate order parameter using much shorter experiments in comparison to the use of
the typical Fourier transform methodology (grey circles). As it will be shown in a later section, this is particularly useful for
complex systems for which only a limited number of indirect dimension points can be measured. Moreover, the fit procedure
enables the estimation of the uncertainty of the value measured (the detailed description for defining the errors is given as SI)
which is more difficult to define with the Fourier transform "read-off" method alone. Figure 4D also shows the result of using
fits with the ideal model (i.e. that does not include the effect of RF inhomogeneity). One observes that these fits become less
accurate with an increase of the number of points used (a comparison with the model including RF inhomogeneity is also given
as SI in Figure S2). This is a consequence of the effect of RF inhomogeneity which becomes more prominent at longer $t_1$ times.
However, Figure 4D also suggests that fitting with the ideal model may provide good results when analysing short $t_1$ intervals.
Based on previous $^2$H NMR measurements, the $\beta$ methylene carbon is expected to have the equivalent order parameters for the
two C−H bonds (Gally et al., 1975). In the next section we demonstrate the usefulness of R-PDLF time-domain fits to analyse
cases of two-component dipolar modulations and estimate the accuracy of the order parameters determined with the proposed
method by comparison with $^2$H NMR quadrupolar splittings.

**Table 1.** Comparison of the order parameter magnitudes, $|S_{CH}|$, determined from the DMPC/DMPCd54 liquid crystalline sample for the $\omega$, $\omega - 1$ and $\omega - 2$ carbons from using the observed splitting in the frequency domain (FT) of a R-PDLF experiment with a total of 32 points in the indirect dimension; from time-domain (TD) fits with numerical simulations either accounting or not for the effect of RF inhomogeneity on R-PDLF experiments and using a single or two order parameters; and from quadrupolar splittings observed with a $^2$H quadrupolar echo.

| Carbon label | FT | Fit with ideal simulations | | Fit with RF inhomogeneity | | $^2$H NMR |
|---|---|---|---|---|---|---|
| | | 1 comp. | 2 comp. | 1 comp. | 2 comp. | |
| $\omega$ | 0.023 | $0.01^{0.014}_{0}$ | $0.028^{0.03}_{0.026}$ | $0.023^{0.024}_{0.023}$ | $0.023^{0.028}_{0.022}$ | 0.023 |
| - | - | - | $0.008^{0.01}_{0.003}$ | - | $0.023^{0.024}_{0.019}$ | - |
| $\omega$-1 | 0.083 | $0.094^{0.097}_{0.092}$ | $0.102^{0.106}_{0.099}$ | $0.090^{0.092}_{0.088}$ | $0.096^{0.103}_{0.088}$ | 0.098 |
| - | - | - | $0.088^{0.091}_{0.084}$ | - | $0.085^{0.092}_{0.081}$ | 0.083 |
| $\omega$-2 | 0.114 | $0.123^{0.125}_{0.12}$ | $0.132^{0.138}_{0.127}$ | $0.118^{0.121}_{0.116}$ | $0.122^{0.132}_{0.115}$ | 0.127 |
| - | - | - | $0.118^{0.121}_{0.114}$ | - | $0.115^{0.122}_{0.11}$ | 0.115 |

## 2.3 Comparison with $^2$H NMR quadrupolar splittings

To further test the accuracy of using a model of R-PDLF experiments that includes RF inhomogeneity, we used a water/DMPC/DMPCd54 liquid crystalline system ($L_\alpha$ phase) such that it was possible to measure the $^2$H NMR spectrum of the perdeuterated acyl chains of the DMPCd54 molecules and compare it with the $^1$H-$^{13}$C dipolar couplings determined from the acyl chains of the molecules with natural abundance of isotopes. Figure 5 shows the R-PDLF dipolar modulation for a number of resolved carbons from the liquid crystalline system and their corresponding fits with and without accounting for RF inhomogeneity. The improvement of the fits by accounting for RF inhomogeneity is visually evident, especially for the smaller coupling. We stress here that the RF inhomogeneity profile used in the simulations was experimentally measured and modelled by a Gaussian profile. The RF inhomogeneity is therefore an objectively quantified a priori constraint that we have obtained from an independent experiment. Moreover, since each of the three carbon peaks analysed is a sum of components from the two individual acyl chains, $sn$-1 and $sn$-2, we also performed the time-domain fits of the dipolar modulations using two components. In the case of the acyl chain methyl groups $\omega$, equivalent dipolar couplings minimize $\chi^2$. On the other hand, for the $\omega - 1$ and $\omega - 2$ methylenes, two distinct dipolar couplings enable a better fit than a single value which suggests a distinct motional geometry in the $sn$-1 and $sn$-2 acyl chains for these methylenes (a detailed analysis is given in the SI). The nonequivalence of these order parameters is in agreement with previous investigations of specifically deuterated DPPC molecules (Seelig and Niederberger, 1974).

The $^2$H NMR spectrum measured from DMPCd54 molecules is shown in Figure 6. The five lowest quadrupolar splittings were used to calculate C–D bond order parameters. The results are included in Table 1 together with the results from the time-domain fits of the dipolar modulations from the molecules with natural abundance of isotopes. The agreement between

the values obtained by using the different techniques is striking. The maximum difference between the order parameter magnitudes determined with the methodology proposed and the values determined from the $^2$H NMR spectrum is below $\pm0.005$, much below the errors reported previously (up to $\pm0.02$) when doing similar comparisons between dipolar recoupling experiments with $^2$H NMR spectroscopy (Gross et al., 1997; Dvinskikh et al., 2005; Ferreira et al., 2013). Here, we used values taken from previous studies for the rigid $^1$H-$^{13}$C dipolar and $^2$H quadrupolar couplings, namely 22 kHz and 126 kHz, respectively (Dvinskikh et al., 2004; Davis et al., 2009). We may therefore exclude a significant isotope effect on the molecular dynamics of the acyl chains in the water/DMPC/DMPCd54 liquid crystalline system. Moreover, we can unambiguously assign the C–D order parameters measured with $^2$H NMR based on the two-component fits of the R-PDLF data. The measurement of the $^2$H NMR spectrum alone (from perdeuterated acyl chains) does not provide information concerning the origin of each of the distinct quadrupolar splittings observed. However, R-PDLF enables chemical shift selectivity in the direct dimension. By fitting the indirect dimension time-domain of the R-PDLF experiment for the $\omega$, $\omega$-1 and $\omega$-2 carbons with two components (one per acyl chain), one readily enables the assignment of the five lowest quadrupole couplings in the $^2$H NMR spectrum.

The agreement between the $^2$H NMR and R-PDLF values highlights the usefulness of modelling R-PDLF data in contrast to the conventional dipolar splitting "read-off" from the frequency-domain analysis. Such distinct dipolar couplings cannot be observed in the dipolar spectrum due to the limited number of points measured as described previously. In fact, in this case, from the "read-off" method one obtains for both the $\omega - 1$ and $\omega - 2$ carbons, the lowest order parameters with great precision. The only fitting parameters used to make all the plots shown in Figure 5 were the dipolar couplings (i.e. order parameters), namely just one fitting parameter in case of a one component fit and two fitting parameters in case of a two-component fit. In the case of the two-component fits we have assumed that the intensity for each component was equal. This is a reasonable assumption in this case, because the expected differences between the order parameters and the dynamics of the two acyl chains are very small. Therefore, the rINEPT intensity of each component should be very similar according to the theoretical model for the dependence of $^{13}$C rINEPT intensity on CH order parameter and correlation times described by Topgaard and coworkers (Nowacka et al., 2010, 2013). In general though, two-component fits may require fitting also the relative magnitude of each component for obtaining a good fit. Here we opted for a minimal amount of fitting parameters, i.e. the simplest model enabling to match the experimental data within experimental uncertainty.

For determining the numbers shown in Table 1, we have additionally used a phenomenological single exponential relaxation contribution as described in the Supplementary Information. This did not affect the order parameters of the two-component fits determined for the carbons $\omega$-1 and $\omega$-2. On the other hand, for the $\omega$ carbon, without inclusion of the single exponential relaxation, the order parameters calculated were 0.026 and 0.020. The line broadening corresponding to the minimum of $\chi^2$ for the single-component fits shown in Table 1 and in Figure 4 using the model that includes RF inhomogeneity was in the range of 4-10 Hz. On the other hand, for the model that does not account for RF inhomogeneity, the resulting line broadening from the minimisation of $\chi^2$ increases significantly in some cases, especially for low frequency dipolar modulations, to as much as 70 Hz. Such an increased relaxation rate is due to the significant deviation of the ideal model from the experimental data as shown in Figure 5 for the $\omega$ carbon. For fits with the ideal model that do not lead to such large relaxation rates, although the minimum of $\chi^2$ is higher than for the model including RF inhomogeneity (as shown in the supplemetary information in

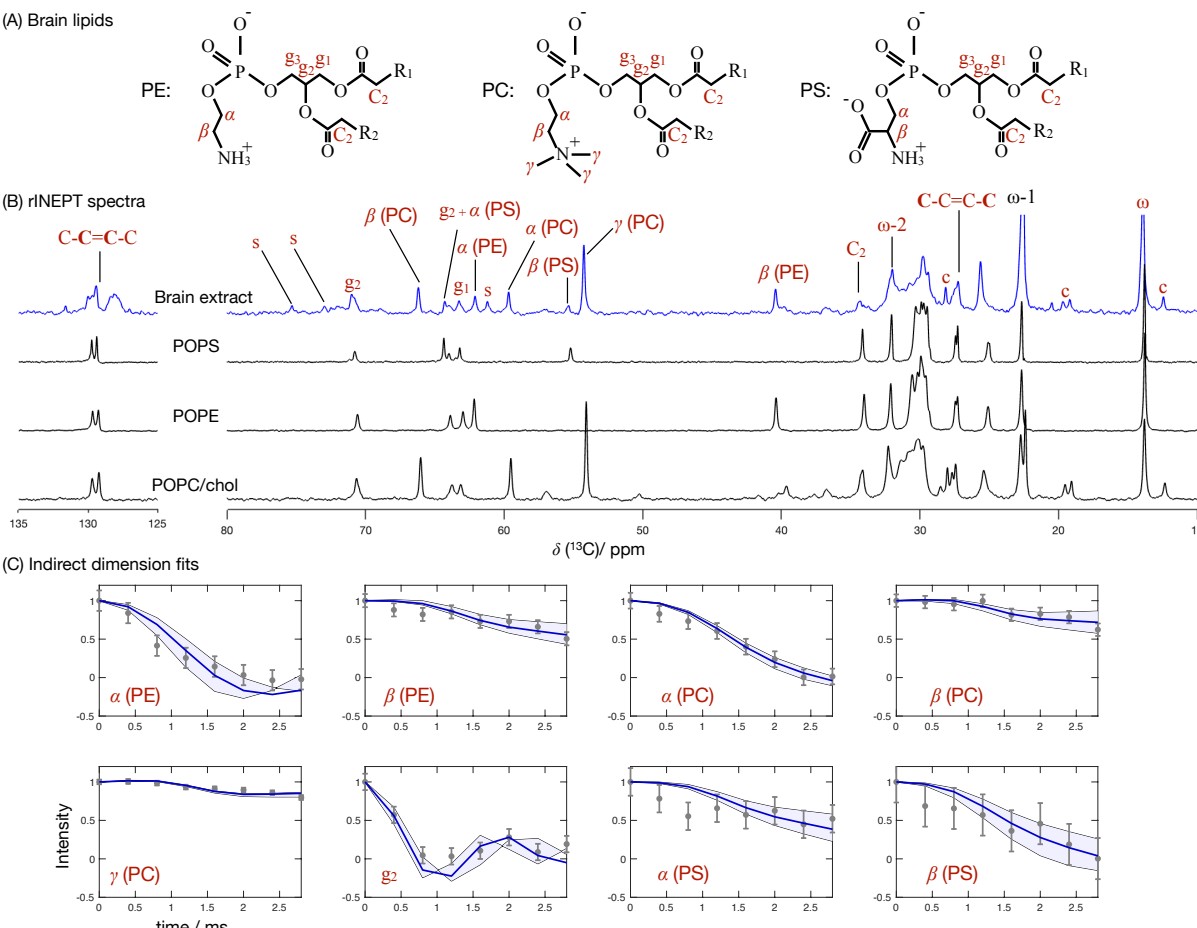

**Figure 7.** Phospholipid headgroup C–H bond order parameter magnitudes determined from a brain lipid extract using time-domain fits accounting for $B_1$ inhomogeneity. The polarization transfer method used was rINEPT, recycle delay was 5 s, MAS rate was 5 kHz and a total of 4096 scans were acquired for each point in the indirect dimension which amounts roughly to 2 days of experimental time. (A) Chemical structures and carbon labels of the most abundant phospholipids in the brain lipid extract: phosphatidylethalonamine (PE), phosphatidylcholine (PC), and phosphatidylserine (PS).(B) $^1$H-$^{13}$C rINEPT spectrum of the brain lipid extract membranes together with reference spectra for the most abundant lipids. The additional labels $c$ and $s$ were used to identify cholesterol and Galactose peaks, respectively. (C) Application of the proposed methodology for fitting the dipolar modulations and determining the corresponding order parameter magnitudes for selected carbons.

Figures S2-S8), the fitting outcome is also good (which is the case for the methylene carbons in Table 1 and for fits of short $t_1$ intervals as shown in Figure 4).

## 2.4 Application to a brain lipid extract

To showcase the applicability of the presented methodology we investigated the molecular structure of a complex membrane system composed of a brain lipid extract with several lipid types present. The lipid mixture consists of the chloroform:methanol extract of porcine brain tissue as purchased from AVANTI LIPIDS.

Figure 7 shows the $^{13}$C spectrum of the brain lipid extract at excess hydration together with spectra acquired from single phospholipid model systems. Due to the much higher complexity of the brain lipid extract sample in comparison to model membrane systems (usually composed of 1-3 distinct components), the number of transients required to achieve a reasonable signal to noise ratio is considerably higher. Here, 4096 transients were acquired for each point in the indirect dimension with a total experimental time cost of approximately 48 hours (in comparison to the 256 transients used for the simple model systems). The comparison of the $^{13}$C spectra measured from the brain extract with the spectra from the simple models enables to identify a number of peaks from different components known to exist in the extract (according to the chemical composition provided by AVANTI LIPIDS for 41.3 wt/wt% of the lipid extract), namely the $\alpha$, $\beta$ and $\gamma$ segments of the headgroup of phosphatidyl-choline (PC), the $\alpha$ and $\beta$ segments of phosphatidylethanolamine (PE), the $\alpha$ and $\beta$ segments of phosphatidylserine (PS). From inspection of the $^{13}$C rINEPT spectrum, and knowing the lipid composition of myelin that makes up most of the dry weight of brain tissue (Norton and Autilio, 1966), it is clear that most of the remaining unknown 58.7 wt/wt% content of the lipid extract is cholesterol plus a considerable fraction of galactose cerebrosides (GalCer) labelled $c$ and $s$ in Figure 7B, respectively.

Figure 7C shows the indirect time-domain modulation for a number of identified peaks in the spectra measured by R-PDLF spectroscopy. In this case, due to the large number of transients required to achieve a reasonable signal to noise ratio (4096 scans), only very short indirect dimension modulations can be measured. Namely, only 8 points in the indirect dimension were acquired for the brain lipid extract sample. Such a reduced total indirect dimension time is not sufficient to obtain dipolar splittings after Fourier transformation. However, application of time-domain fitting including RF inhomogeneity enables to determine the dipolar couplings and therefore estimate C–H bond order parameters. The estimated order parameters are presented in Table 2.

The headgroup order parameters determined from the set of different phospholipid types in the brain lipid extract are very much in line with what is observed in corresponding lipid model membranes. The only exception is the PS $\beta$ order parameter with a rather small order parameter equal to $0.04^{0.052}_{0.031}$. This suggests that the brain lipid extract may contain calcium ions that are known to significantly affect the PS headgroup orientation. A thorough analysis of the order parameters in terms of the molecular structure in the complex brain lipid extract system will not be given here. Such analysis should be done in combination with experiments and MD simulations on PC/PE/PS/cholesterol/GalCer model membranes which resemble the lipid extract composition, to investigate potential phase separation behavior, effects of distinct ions present in the system and pH, or ultimately the effect of macromolecules such as basic myelin protein, the most abundant protein in myelin. We believe that such a combined approach will lead to a more accurate description of the lipid/protein interactions in myelin, which may be important in the context of a number of diseases related to demyelination. The methodology proposed here will enable to perform such investigations with higher accuracy using shorter experiments and without the requirement of isotopic labelling.

**Table 2.** Order parameter magnitudes determined from a brain lipid extract using the methodology proposed in this work as shown in Figure 7. The superscript and subscript for each value denote upper and lower 90% confidence bounds as shown in the Supplementary Information (Figure S9). The values are compared to $^2$H NMR values measured from phospholipid model systems reported previously. [a] $T = 40°C$, [b] $T = 30°C$ and [c] $T = 23°C$

| Carbon segment | Brain extract $|S_{CH}|$ | Model system $|S_{CD}|$ | model |
|---|---|---|---|
| $\alpha$ (PC) | $0.044^{0.048}_{0.041}$ | $0.048^a$ | DPPC/chol (Brown and Seelig, 1978) |
| | | $0.049^b$ | DOPC (Ulrich and Watts, 1994) |
| | | $0.048^c$ | POPC (Bechinger and Seelig, 1991) |
| $\beta$ (PC) | $0.014^{0.018}_{0}$ | $0.024^a$ | DPPC/chol (Brown and Seelig, 1978) |
| | | $0.029^b$ | DOPC (Ulrich and Watts, 1994) |
| | | $0.044^c$ | POPC (Bechinger and Seelig, 1991) |
| $\gamma$ (PC) | $0.004^{0.008}_{0}$ | $0.008^a$ | DPPC/chol (Brown and Seelig, 1978) |
| $\alpha$ (PE) | $0.066^{0.082}_{0.055}$ | $0.063^a$ | DPPE/chol (Brown and Seelig, 1978) |
| $\beta$ (PE) | $0.019^{0.024}_{0.015}$ | $0.020^a$ | DPPE/chol (Brown and Seelig, 1978) |
| $\alpha$ (PS) | $0.026^{0.032}_{0.018}$ | $0.016^a$ | POPC/POPS (Roux and Bloom, 1990) |
| $\beta$ (PS) | $0.04^{0.052}_{0.031}$ | $0.091^a$ | POPC/POPS (Roux and Bloom, 1990) |
| | | $0.062^a$ | + 1 M CaCl$_2$ (Roux and Bloom, 1990) |

Moreover, we envision applications to many other complex membranes such as bacterial membrane models, eukaryotic membrane models with cell type or organelle specific compositions, technological systems such as lipid-nanoparticles, as well as to address the molecular structure/orientation of drugs, peptides or other molecules with anisotropic motion in lipid membranes.

# 3    Conclusions

In summary, we have described a PDLF NMR methodology that consists of performing time-domain fits of the dipolar modulation with numerical simulations that account for the RF spatial inhomogeneity of the probe used. The RF inhomogeneity is experimentally measured and is an a priori constraint. A dipolar modulation of a single component is therefore fitted with only two parameters, the dipolar coupling and a phenomenological relaxation time to account for exponential relaxation. The proposed methodology enables one to determine C–H bond order parameters from simple model systems with a much higher accuracy than previously and enables to investigate complex lipid membrane systems that were so far inaccessible using the conventional "read-off" PDLF NMR methodology. We believe that the method presented will be extremely useful in the future concerning molecular structural investigations of complex systems such as multi-component models, lipid extracts and lipid membranes with drugs, peptides or other molecules incorporated. Moreover, the higher accuracy and farther reach of the method will also be fundamental to validate molecular dynamics simulations for which structural experimental data was not accessible so far.

*Code and data availability.*    https://github.com/tfmFerreira/inhomogeneous-rf-nmr.git

*Author contributions.*    T.M.F. idealized the project. T.M.F. and A.W. developed the code. T.M.F and A.W. performed experiments and processed the experimental data. T.M.F., A.W. and K.S. analysed and discussed the results and manuscript. T.M.F. wrote the manuscript.

*Competing interests.*    no competing interests

*Acknowledgements.*    T.M.F. greatly acknowledges Alexey Krushelnitsky for invaluable support and discussions. This research study was funded by the German Research Foundation (Deutsche Forschungsgemeinschaft,DFG) [project number 189853844, TRR 102 (T.M.F. and A.W.)].

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
