# Peer review of "Time-domain R-PDLF NMR for molecular structure determination in complex lipid membranes"

_Magnetic Resonance, 2022_

## Author Response (AR1)

**General comments to the Reviewers**

We thank the reviewers for their suggestions and critical evaluation of the manuscript. Below we address all the points raised by the reviewers and describe our intended changes for revising the manuscript. First we give a general description of what we believe to be the novelty of the work presented to use as a reference for the replies to the specific comments.

The **major novelty** described is the **use of NMR simulations including the effect of RF inhomogeneity to evaluate the indirect time-domain (TD) signal measured in R-PDLF experiments**. This **enables to determine order parameters from more complex samples** than the ones that have been investigated previously.

R-PDLF spectroscopy has been so far the main proton detected local field (PDLF) NMR method to obtain lipid C-H order parameters from lipid bilayers, especially in the context of validation of all-atom MD simulations of lipid bilayers. The advantage of PDLF methods over non-PDLF methods is that, in PDLF experiments, the Hamiltonian during the indirect dimension is effectively a sum of spin-pair interactions, while for non-PDLF (such as REDOR and DIPSHIFT) the Hamiltonian refers to a multi-spin network. This means that the C-H order parameters determined from PDLF experiments are therefore directly comparable to C-H order parameters measured with $^2$H NMR. This is not necessarily the case for non-PDLF experiments. $^2$H NMR enables the most accurate C-H order parameter determination but isotopic labelling is required and there is no chemical shift selectivity. The main two advantages of PDLF NMR over $^2$H NMR are: (1) its applicability to samples with a natural abundance of isotopes and (2) providing chemical shift selectivity. Contrarily to $^2$H NMR, PDLF NMR is therefore potentially applicable to complex biological samples.

The **conventional approach** to analyse PDLF experiments has been to perform a 2D Fourier transform of the measured data and to determine the order parameters directly from the dipolar splittings in the indirect dimension (**without any fitting procedure**). In cases of short time intervals covered in the indirect dimension - which is often a necessity when dealing with complex samples at natural abundance of isotopes because of the high number of transients required in the direct dimension - the FT procedure does not yield any splitting and the determination of the dipolar coupling is only possible by fitting the data with a model. Contrarily to what has been done with non-PDLF NMR experiments like REDOR or DIPSHIFT, the fitting procedure of PDLF experiments has not been recurrently done and ideal numerical simulations do not enable good fits (as it is shown in the manuscript, Figure 5C) specially for the smaller couplings measured from lipids. **We demonstrate that by accounting for RF field inhomogeneity in NMR simulations, the R-PDLF data can be fitted perfectly and that highly accurate order parameters can be determined from short data i.e. from data that would not yield splittings after the 2D NMR FT procedure (Figure 3D) due to the limited number of points measured.**

Additionally, to best of our knowledge, it is the first time that C-H order parameters from $^2$H NMR and PDLF NMR are measured from the same sample and compared (Figure 5 and Table I). This comparison indicates that the precision of order parameters determined from peaks that consist of two components can be made higher using the fitting procedure suggested in comparison to the conventional approach.

The fitting method suggested is then implemented on a complex lipid membrane with multiple components.

We believe that the procedure shown is highly relevant for future studies of more complex membranes. We are using it already in distinct projects that are being currently undertaken in our group. This publication will serve as a reference for these studies and will help other groups interested in lipid membranes to implement such a strategy (we made all the simulations and scripts to generate simulations publicly available). The findings shown are also most possibly relevant also for investigations of peptides and membrane proteins inserted in lipid membranes.

**Replies to specific comments**

**Review #1**

General comments:

The manuscript by Wurl et al. describes a maximum likelihood approach to fitting data from R-type 1H-detected local field experiments in which simulated data computed by forward-modeling, including empirically-determined field inhomogeneity. While this is a promising approach in principle, the method described lacks an effort at error analysis or propagation other than cross-validation against results from 2H experiments. There is also no discussion of model order selection. These deficits undermine the support fot the claims that the method yields better accuracy. **In addition to performing error analysis, the authors should attend to poor English usage in the manuscript**.

Specific comments:

The comment that time-domain fitting avoids the problems of short data records is incorrect, as error analysis would demonstrate (the errors will increase as the data records become shorter). The discrete Fourier transform is, after all, also a maximum likelihood method in which the model is a Fourier series. The time domain fitting approach here works better because it has fewer degrees of freedom (so long as the assumptions are valid, e.g. the field inhomogeneity). Apropos degrees of freedom, the authors do not consider the problem of model order selection – e.g. one vs. two components. A two component model will always fit the empirical data better. The question is whether the better fit is statistically significant in light of the increase of degrees of freedom. This can be approached using the minimum description length or other measures of information content.

While various methods for error analysis could be applied to the maximum likelihood approach described by the authors (e.g. Cramer-Rao), a more rigorous approach would be to use Bayesian analysis, in which the field inhomogeneity is treated as a prior probability. This would allow the impact of the uncertainty in the determination of the inhomogeneity to be propagated to the posterior probability distribution. It would be a natural extension of the maximum likelihood approach described here.

**Replies to comments:**

We believe that reviewer #1 puts forward rather general arguments that do not acknowledge the methodological aspects and the experimental details of our work, which are specific in terms of applying the R-PDLF method to obtain lipid CH order parameters. We note that is relevant to consider our arguments presented above, namely that the PDLF concept implies *a priori* that fits can be done on the basis of either single couplings or inhomogeneous distributions thereof (bimodal, certain distribution shapes, etc.). In lipid systems, it is well established that CH order parameters are well-defined and single-valued, as confirmed by previous work in the field as well as our analyses and comparisons. RF inhomogeneity is not an uncertainty subject to statistical treatment, it is an objective experimental fact, where our improved fits (having no additional parameter, only a single coupling in either case) clearly demonstrate that it is the main origin of systematic deviations between data and fits (beyond noise). In a few special cases (spectral signals), a bimodal fit describes the TD data significantly better; a fact supported independently by $^2$H NMR results. It is not our purpose to evaluate the possibility of R-PDLF-based distribution analysis (e.g. by regularization); this would be interesting on its own right, but is beyond the scope of experiments on lipid systems that are known to be homogeneous. We merely show that an unprecedented accuracy can be reached by a well-justified data analysis.

"While this is a promising approach in principle, the method described lacks an effort at error analysis or propagation other than cross-validation against results from 2H experiments. There is also no discussion of model order selection. These deficits undermine the support fot the claims that the method yields better accuracy."

The statement that we did not conduct an error analysis or error propagation is not correct. We display this analysis clearly in Figure 3D where we show the outcome of using the TD fit analysis including RF inhomogeneity in comparison to what one gets from using the conventional approach (FT to obtain a dipolar splitting without any fitting). In this figure, we present error bars from the fitting evaluation. The error bars decrease as the amount of data fitted increases. This figure shows clearly that the determination of the order parameter from the TD fitting yields precise values from much shorter data sets in comparison to the conventional approach.

We do not consider any *model order selection* because the model is always the same: spin-pair simulations of the R-PDLF NMR experiment including RF inhomogeneity. We only fit two-components in cases for which we know that there exist two-components (two distinct acyl chains). Please note that this is not an assumption. We have now tried to clarify this in the text (lines 184-189 and 208-219 in diff.pdf).

Moreover, we believe that the best proof that the method works is precisely the comparison with $^2$H NMR (the most accurate method to determine C-H order parameters). This is the strongest support for claiming the high accuracy of the method.

"The comment that time-domain fitting avoids the problems of short data records is incorrect, as error analysis would demonstrate (the errors will increase as the data records become shorter). The discrete Fourier transform is, after all, also a maximum likelihood method in which the model is a Fourier series."

We disagree with this statement, and point to reviewer #2, who correctly states that the FT is a linear operation that does not change the information content inherent to the data. We again refer to figure 3D (and to the previous reply) where it is clearly shown that a TD fit enables to extract information from shorter data (indeed with an increase of error towards shorter data) for which FT processing does not yield any dipolar splitting. Taking into account also the comments from reviewer #2, we should maybe clarify that **what we claim is not that a TD fit is more accurate than a frequency domain (FD) fit**. It does not matter whether a fitting procedure (which we claim is the relevant improvement over the existing approach of "reading off" splittings) is applied to TD or FD data. But "reading off" a splitting from the FD is impossible when the TD contains too few points. While being a trivial and well-known fact, it shows that a quantitative error analysis of the FT in terms of being a "maximum likelihood method" is not going to provide relevant insights. What we compare is the outcome of performing a TD fit (using RF inhomogeneity) with the outcome of using the conventional approach i.e. using only FFT processing of the TD data to determine a dipolar splitting (the conventional approach, without any fit of the data involved). We have now tried to emphasise this in the abstract and throughout the text (lines 65-70 and 154-157 in diff.pdf).

While various methods for error analysis could be applied to the maximum likelihood approach described by the authors (e.g. Cramer-Rao), a more rigorous approach would be to use Bayesian analysis, in which the field inhomogeneity is treated as a prior probability. This would allow the impact of the uncertainty in the determination of the inhomogeneity to be propagated to the posterior probability distribution. It would be a natural extension of the maximum likelihood approach described here.

It is very important to note here that the RF inhomogeneity is not unknown information when the fit is performed but rather an **objectively quantified *a priori* constraint** that we have obtained from an independent experiment as described in the Supplementary information (see e.g. Figure S1). Nevertheless, as suggested by the reviewer, we have investigated how the fitted values are affected by the gaussian width that describes the RF inhomogeneity. These values do not change within a range of gaussian functions that is clearlybeyond the uncertainty of the RF inhomogeneity measurement. We refrain from using the error analysis procedures suggested since we believe that the comparison with $^2$H NMR and the simple error analysis already presented should be sufficient in the scope of the presented findings. In the new version we stress that the RF inhomogeneity is *a priori* experimentally measured information used to generate the fitting model (lines 181-184 and 272-274 in diff.pdf).

"**In addition to performing error analysis, the authors should attend to poor English usage in the manuscript**."

We have tried to improve the written quality of the manuscript the best we could, prioritising the clarity of the text. It would help if the reviewer would give more detailed information concerning this comment.

**Review #2**

Wurl et al. report fitting of recoupled dipolar oscillations in the time domain and take into consideration the RF inhomogeneity of the probe. Measurement of the RF field inhomogeneity using a gradient, though not new, may be useful. I see several major issues with the manuscript. 1st, time domain fitting of dipolar recoupled spectra is not new. See for example the extensive work with DIPSHIFT, REDOR, including fitting of multiple distances (see for example J. Phys. Chem. B 2007, 111, 27, 7802–7811). 2nd, from a theoretical standpoint, the authors do not explain why time domain analysis is better than frequency domain analysis; since the FT is a linear operation, it should not result in any loss of information. Each point from the time domain produces a 'basis function' of multiple points in the frequency domain, such that it seems fitting either should be possible. The authors did not make a direct comparison between time domain and frequency domain fitting, but showed only comparisons of the freq. domain data without fitting and sometimes without even considering RF inhomogeneity, though that part is not always clear. 3rd, consideration of RF inhomogeneity in time domain fitting of similar recoupled dipolar interactions has been reported before (e.g. in J. Phys. Chem. Lett. 2022, 13, 18–24). I think that if the authors take the above into consideration, that the article could still be helpful for those working in the more dynamic application areas, such as lipid properties.

More minor things:

In the abstract, R-type is rather colloquial. 'R symmetry sequence' or similar would be more precise.

Perhaps there is a review article that could be helpful in the introduction (first paragraph)?

Duplicate citation line 34

Line 49: a strategy to develop a strategy …

Line 52: more recent work in this area is: J. Chem. Phys. **150**, 134201 (2019); https://doi.org/10.1063/1.5088100 and J Magn Reson 2020 Oct;319:106827. doi: 10.1016/j.jmr.2020.106827

Line 59, the authors probably mean rather a reduction in the maximum time in the indirect dimension.

Line 69. Considering comments above, mention of the exceptions to this statement are relevant here.

Line 101. kHz would seem to be the more natural unit (at what magnetic field(s) is this statement relevant).

Fig. 2. values in ppm are probably not meaningful without knowing the magnetic field. values in Hz/kHz would be more important. Indication of units for the offset missing. Which

R sequence was used? Ok, it is mentioned in Fig. 1, but the choice of R sequence could be discussed somewhere.

Line 106. more details would be helpful in main text. this is critical to the story since application of this Gradient method seems to be the main innovation. In SI, is the sample static? is Z-gradient better than X or XZ?

Fig. 3. what were the fit parameters? Amplitude (fraction of each species) or only couplings? Why are the FTs of the red and blue curves of panel D not shown?

Fig 4. did both the time domain as well as freq. domain fits include the same treatment of RF inhomogeneity? How does the fit differ if done in the freq. domain (with all other parameters the same)?

Line 171. How were assignments done unambiguously?

Line 193. 'considerably higher' what is considerable? Quantification here would be good.

Line 194. Label of gamma missing in fig 1A.

Line 218. 'higher accuracy' compared to what ? 2H accuracy is not compared (just references to literature values which may have been acquired somewhat differently, and error bars which represent precision).

Fig S7 bottom left. two axes labeled the same. they should be Indexed. panel C explain the labels u and i (maybe that is an L?). also applies to other supplement figures.

Gaussian: capitalize

**Replies to comments:**

"Measurement of the RF field inhomogeneity using a gradient, though not new, may be useful. I see several major issues with the manuscript. 1st, time domain fitting of dipolar recoupled spectra is not new. See for example the extensive work with DIPSHIFT, REDOR, including fitting of multiple distances (see for example J. Phys. Chem. B 2007, 111, 27, 7802–7811)."

We do not claim that time-domain fitting is new. Obviously, REDOR and DIPSHIFT are based on time-domain fits. We tried to clarify this in the text (lines 65-70 in diff.pdf). What we address is the fitting of R-PDLF experiments, which provide C-H bond resolution, unlike the non-PDLF methods referred. Some attempts for fitting R-PDLF experiments were done previously but without considering RF inhomogeneity. Here we show that accounting for RF inhomogeneity in the simulations is crucial to get accurate fits of the data (e.g. Figure 5).

2nd, from a theoretical standpoint, the authors do not explain why time domain analysis is better than frequency domain analysis; since the FT is a linear operation, it should not result in any loss of information. Each point from the time domain produces a 'basis function' of multiple points in the frequency domain, such that it seems fitting either should be possible. The authors did not make a direct comparison between time domain and frequency domain fitting, but showed only comparisons of the freq. domain data without fitting and sometimes without even considering RF inhomogeneity, though that part is not always clear.

We again clarify here that **the comparison is not between obtaining dipolar couplings from fits of the TD domain *versus* fits of the FT domain, but rather between fits of the TD domain and by using simply the value of the dipolar splitting observed in the dipolar dimension of the experimental spectrum (with no fit involved)**, where the

latter has been customary in the literature. We do not see any advantage in using an FT at all, as the actual data points in both the simulations and the experiments are measured in the TD. So there is no reason to invest any extra numerical effort. Please see also reply to reviewer #1 above. We have now highlighted this in the abstract and throughout the text (lines 65-70 and 154-157 in diff.pdf).

3rd, consideration of RF inhomogeneity in time domain fitting of similar recoupled dipolar interactions has been reported before (e.g. in J. Phys. Chem. Lett. 2022, 13, 18–24). I think that if the authors take the above into consideration, that the article could still be helpful for those working in the more dynamic application areas, such as lipid properties.

Indeed, the use of RF inhomogeneity was also used in this work for a distinct dipolar recoupling methodology that was applied to protein structure determination. We were not aware of this work and will mention it and include it in our references (lines 75-76 in diff.pdf). This method is based on CP transfer which does not provide the C-H bond resolution as in the R-PDLF experiment with rINEPT as polarisation transfer. The mentioned methodology was never used in the context of lipid membranes where small dipolar couplings are expected and the feasibility of its application in this context is not clear. Therefore we still believe that it is useful to show our findings concerning the R-PDLF sequence which is the method that has provided most of the dipolar recouping data for lipid membrane systems.

In the abstract, R-type is rather colloquial. 'R symmetry sequence' or similar would be more precise.

We changed this as suggested.

Perhaps there is a review article that could be helpful in the introduction (first paragraph)?

We have included two more review references (lines 25-26 in diff.pdf)

Duplicate citation line 34

Thanks.

Line 49: a strategy to develop a strategy …

We changed "strategies" to "methods".

Line 52: more recent work in this area is: J. Chem. Phys. **150**, 134201 (2019); https://doi.org/10.1063/1.5088100 and J Magn Reson 2020 Oct;319:106827. doi: 10.1016/j.jmr.2020.106827

These references are now also included.

Line 59, the authors probably mean rather a reduction in the maximum time in the indirect dimension.

(Probably line 54 instead of 59?) Yes, we changed it to the suggested sentence formulation.

Line 69. Considering comments above, mention of the exceptions to this statement are relevant here.

We could not understand this comment. The sentence explicitly refers to PDLF techniques (so REDOR and DIPSHIFT do not apply).

Line 101. kHz would seem to be the more natural unit (at what magnetic field(s) is this statement relevant).

We included the Larmor frequency for protons used (400 MHz)

Fig. 2. values in ppm are probably not meaningful without knowing the magnetic field. values in Hz/kHz would be more important. Indication of units for the offset missing.

We have now included this information in the figure caption.

Which R sequence was used? Ok, it is mentioned in Fig. 1, but the choice of R sequence could be discussed somewhere.

We simply used the R sequence used in the original paper for the R-PDLF experiment. This is the only R sequence that has been used for measuring lipid order parameters previously. We have mentioned this in the introduction (lines 46-48 in diff.pdf) .

Line 106. more details would be helpful in main text. this is critical to the story since application of this Gradient method seems to be the main innovation. In SI, is the sample static? is Z-gradient better than X or XZ?

We simply used the method suggested by Odedra, S. and Wimperis, S., J. Mag. Reson., 231, 95 – 99, https://doi.org/https://doi.org/10.1016/j.jmr.2013.04.002, 2013. The feasibility of using the gradient over X depends on the relative orientation of the X-gradient with the rotor long axis. We used a simple approach. The linear gradient should be ideally along the rotor axis. However, the agreement with 2H NMR values and FT splittings at long indirect times shows that this approximation is sufficient.

Fig. 3. what were the fit parameters? Amplitude (fraction of each species) or only couplings? Why are the FTs of the red and blue curves of panel D not shown?

The fitting parameters are only the couplings. No special reason. We have included them now. Note that this changed the fitted values because the random noise added to make this plot was generated once again.

Fig 4. did both the time domain as well as freq. domain fits include the same treatment of RF inhomogeneity? How does the fit differ if done in the freq. domain (with all other parameters the same)?

There are no frequency domain fits here, only the determination of the splitting from the FFT of the time domain data. If the fit would be done in the frequency domain using the FT of the simulated data with RF inhomogeneity, the obtained values are possibly the same as for the TD domain fits. We have not tried this. Fitting the frequency domain instead of the TD domain would required additional processing steps so we have not considered doing this. We added a few sentences concerning this (lines 65-70 and 154-157 in diff.pdf)

Line 171. How were assignments done unambiguously?

The point here is the following. The measurement of the 2H NMR spectrum alone does not provide information concerning the origin of each distinct quadrupolar splitting observed. However, R-PDLF enables chemical shift selectivity in the direct dimension. By fitting the

indirect dimension time domain of the R-PDLF experiment for the ω, ω-1 and ω-2 carbons with two components, we readily enable the identification of the five lowest quadrupole couplings in the 2H NMR spectrum. We tried to clarify this in the text (lines 201-205 in diff.pdf).

Line 193. 'considerably higher' what is considerable? Quantification here would be good.

The number of transients (4096) was given in figure 7. This compares with around 256 scans for the simple model systems. We have now made this clear in the text (lines 237-239 in diff.pdf).

Line 194. Label of gamma missing in fig 1A.

Thanks.

Line 218. 'higher accuracy' compared to what ? 2H accuracy is not compared (just references to literature values which may have been acquired somewhat differently, and error bars which represent precision).

In comparison to simply calculating the order parameter from the dipolar splitting observed in the spectrum (i.e. with no fitting). We have now clarified this in the sentence.

Fig S7 bottom left. two axes labeled the same. they should be Indexed. panel C explain the labels u and i (maybe that is an L?). also applies to other supplement figures.

Thanks. We did now. Indeed it is an u (upper) and l (lower).

Gaussian: capitalize

Thanks.

We thank both reviewers for their critical evaluation, specially reviewer #2 for the very detailed proof reading.

Sincerely,
Tiago Mendes Ferreira
(on behalf of all the co-authors)

---

## Author Response (AR2)

Dear Editor and reviewers,

Thank you for your critical assessment of our manuscript. Below we provide replies to your comments, and describe all the minor changes done as requested. We also included a few other minor changes that, although not requested, improve, in our view, the final version of the manuscript.

Editor comments:

The Proton Detected Local Field Experiment is used, together with a modified data evaluation scheme (including considering the effect of the rf-inhomogeneity) to determine order parameters. Based on the comments of referee 1 ("In particular my main concern was the claim that time domain analysis is better than freq. domain analysis. After reading the letter, it is clear that this was not the intended claim. However, there is still once instance where I think the phrasing is problematic in this regard. page 4 line 112:'Time-domain analysis can be used to circumvent the limitations outlined above entirely'. Rather, considering RF inhomogeneity is important and time domain analysis is convenient/ efficient." ) and ,my own reading, the manuscript needs a further revision to make it clear that the time-domain fitting is more CONVENIENT but does not provide more information or needs less measurement time) than a frequency-domain FIT based on the same model (including rf inhomogeneity). The comparison with a by eye inspection of the Fourier transform is not an adequate comparison and falsely suggests that the time-domain method is superior. also, the time domain method does not need shorter rf irradiation periods as claimed.

We have tried to clarify that there is no advantage of using TD fits over FD fits. We have introduced for instance the following new sentences:

- (in Abstract) "In comparison to the analysis of dipolar splittings without any fitting procedure, the accurate modelling of PDLF measurements makes possible…"

- "Analysing the measured data with a fitting model (in either the time or frequency domains) can be used to circumvent the limitations outlined above"

We stress again that the comparison of TD fits with FD fits was never our intention, as also emphasised by reviewer 1, and we were actually surprised for this issue to come up. Our intention was to compare the result of fitting a model with the result of "reading off" the dipolar splitting from the spectrum (with no fitting) since the latter approach is often used including by us. Therefore, when we state that the methodology enables to use shorter experiments, this is in connection to the "read off" method that requires more points in the indirect dimension for determining a dipolar coupling.

Furthermore, as suggested by referee 1, the hype should be tuned down and words like "unprecedented level of detail" and the extensive use of new or novel (in particular in the abstract) should be avoided.

We have tried to remove these instances and tried to cool down our excitement over the proposed strategy.

Reviewer 1

The second submission of this article is substantially improved, addressing the points raised in review. In particular my main concern was the claim that time domain analysis is better than freq. domain analysis. After reading the letter, it is clear that this was not the intended claim. However, there is still once instance where I think the phrasing is problematic in this regard. page 4 line 112:'Time-domain analysis can be used to circumvent the limitations outlined above entirely'. Rather, considering RF inhomogeneity is important and time domain analysis is convenient/efficient.

We have now replaced the sentence "Time-domain analysis can be used to circumvent the limitations outlined above entirely" by "Analysing the measured data with a fitting model (in either the time or frequency domains) can be used to circumvent the limitations outlined above".

The claim of improved fitting accuracy is addressed via synthetic data adequately. It might also be interesting to show fits that do not consider RF inhomogeneity in Table 1. This could further strengthen the argument since the current presentation comparing with 2H data does not quantify the improvement.

We thank the reviewer for this suggestion. We now included the result of the fits performed without considering RF inhomogeneity (the ideal model) in table 1 and in Figure 4, and show the minimum chi^2 for these fits in the supplementary information. We also briefly discussed this new information in the text.

'new' and 'new method' is overused in the abstract for my taste, but fine if it fits the style of the journal.

We agree about the overuse and tried to reduce the use of these instances throughout the text.

These suggested changes are so minor that I do not see a need to review another version.

Reviewer 2

The authors' responses to concerns raised are mainly deflection rather than substance. Some of the responses strain credulity, and appear to reflect fundamental misunderstanding of the nature of inverse problems and the role of model fitting.

For example,

"RF inhomogeneity is not an uncertainty subject to statistical treatment, it is an objective experimental fact..."
"We do not consider any model order selection because the model is always the same: spin-pair simulations of the R-PDLF NMR experiment including RF inhomogeneity. We only fit two-components in cases for which we know that there exist two-components (two distinct acyl chains). Please note that this is not an assumption. "
"... a TD fit enables to extract information from shorter data (indeed with an increase of error towards shorter data) for which FT processing does not yield any dipolar splitting."
The authors might consult a book such as Körner "Fourier Analysis". It reprints an essay by Haldane on error analysis.

While the second reviewer is to be commended for the extensive editing, it is not the responsibility of reviewers to edit manuscripts for grammar or syntax.

We have tried to optimise further our error analysis which was the concern of reviewer 2 in his first report. Namely, we now used $\chi2$ boundaries for defining our maximum and minimum confidence bounds as described in detail in the supplementary information.

List of the changes made:

- We tried to remove words like new and novel as suggested.
- Figure 4D was updated to include results from the ideal model.
- Table 1 was updated with results from the ideal model including a brief discussion of these results in the text.
- The experimental errors were modified to correspond to the standard deviation instead of the previous maximum error (which were used previously). For the confidence bounds, $\chi2$ is used now as described in detail in the SI.
- Figure 7 in the main text was updated according to the preceding item.
- The figures S2 to S8 were modified in accordance to the more simple and standard analysis of fitting errors used now.
- A figure with the $\chi2$ dependence for the analysis of myelin carbons was added to the supplementary information (Figure S9).
- Table 1 and Table 2 were updated with the new fitting error estimations.
- The last paragraph of section 2.3 was removed since it was incorrectly stated that POPE membranes gave a higher line broadening than the DMPC/DMPCd membranes. This was a mistake as could be seen in the previous SI plots for these samples.

Again, thanks to all Reviewers and Editor for their evaluation of our manuscript.

Sincerely,
Tiago Mendes Ferreira,
On behalf of all the co-authors.